
# BVOC-aerosol-climate feedbacks investigated using NorESM

Moa K. Sporre[1], Sara M. Blichner[1], Inger H. H. Karset[1], Risto Makkonen[2,3], and Terje K. Berntsen[1,4]

[1]Department of Geosciences, University of Oslo, Oslo, 0315, Norway
[2]Institute for Atmospheric and Earth System Research / Physics, Faculty of Science, P.O. Box 64, FI-00014, University of Helsinki, Finland
[3]Climate System Research, Finnish Meteorological Institute, P.O. Box 503, FI-00101, Helsinki, Finland
[4]CICERO Center for International Climate Research, Oslo, Norway

**Correspondence:** Moa Sporre (m.k.sporre@geo.uio.no)

**Abstract.** Both higher temperatures and increased $CO_2$ concentrations are (separately) expected to increase the emissions of biogenic volatile organic compounds (BVOCs). This has been proposed to initiate negative climate feedback mechanisms through increased formation of secondary organic aerosol (SOA). More SOA can make the clouds more reflective, which can provide a cooling. Furthermore, the increase in SOA formation has also been proposed to lead to increased aerosol scattering, resulting in an increase in diffuse radiation. This could boost gross primary production (GPP) and further increase BVOC emissions. In this study, we have used the Norwegian Earth System Model (NorESM) to investigate both these feedback mechanisms. Three sets of experiments were set up to quantify the feedback w.r.t. (1) doubling the $CO_2$, (2) increasing temperatures corresponding to a doubling of $CO_2$ and (3) the combined effect of both doubling $CO_2$ and a warmer climate. For each of these experiments we ran two simulations, with identical set-up, except for the BVOC emissions. One simulation was run with interactive BVOC emissions, allowing the BVOC emissions to respond to changes in $CO_2$ and/or climate. In the other simulation, the BVOC emissions were fixed at present day conditions, essentially turning the feedback off. The comparison of these two simulations enable us to investigate each step along the feedback as well as estimate their overall relevance for the future climate.

We find that the BVOC feedback can have a significant impact on the climate. The annual global BVOC emissions are up to 63 % higher when the feedback is turned on compared to when the feedback is turned off, with the largest response when both $CO_2$ and climate are changed. The higher BVOC levels lead to the formation of more SOA mass (max 53 %), and result in more particles through increased new particle formation as well as larger particles through increased condensation. The corresponding changes in the cloud properties lead to a -0.43 W m$^{-2}$ stronger net cloud forcing. This effect becomes about 50 % stronger when the model is run with reduced anthropogenic aerosol emissions, indicating that the feedback will become even more important as we decrease aerosol and precursor emissions. We do not find boost in GPP due to increased aerosol scattering on a global scale. Instead, the fate of the GPP seem to be controlled by the BVOC effects on the clouds. However, the higher aerosol scattering associated with the higher BVOC emissions is found to also contribute with a potentially important enhanced negative direct forcing (-0.06 W m$^{-2}$). The global total aerosol forcing associated with the feedback is -0.49 W m$^{-2}$ indicating that it has the potential to offset about 13 % of the forcing associated with a doubling of $CO_2$.





# 1 Introduction

Our climate is warming due to rising atmospheric levels of greenhouse gases originating from human activities (IPCC, 2013). Feedback mechanisms that arise from increasing temperatures and/or greenhouse gas concentrations can enhance or dampen the temperature increase, and contribute to the overall uncertainty in predicting the future climate. Increased emissions of

biogenic volatile organic compounds (BVOC) from terrestrial vegetation caused by increasing temperature and $CO_2$ levels has been proposed to induce a negative climate feedback (Kulmala et al., 2004, 2013). Higher BVOC concentrations results in higher aerosol number and mass concentration, which cool the climate by inducing changes in cloud properties (Twomey, 1974; Albrecht, 1989). Aerosol particles and their interactions with clouds and climate constitutes one of the largest uncertainties in assessing our future climate (IPCC, 2013).

BVOCs are important sources of aerosol particles (Glasius and Goldstein, 2016), especially in pristine forest regions (Tunved et al., 2006). The most important BVOC compounds for aerosol formation are isoprene, monoterpenes and sesquiterpenes (Kulmala et al., 2013), and their emissions have been estimated to 700-1000 TgC annually (Laothawornkitkul et al., 2009). Through oxidation in the atmosphere, these compounds become less volatile and may contribute to aerosol formation. The main oxidation agents are OH, $O_3$ and $NO_3$ radicals (Shrivastava et al., 2017). The oxidation products from monoterpenes

have been found to be particularly important for new particle formation, while the oxidation products from isoprene have been found to predominantly participate in condensation onto pre-existing aerosols (Jokinen et al., 2015). How sensitive the aerosol number concentration are to changes in BVOC emissions depend on the anthropogenic and natural aerosol load. It has been shown that the BVOC had greater influence on the number and mass concentration in the pre-industrial (PI) atmosphere (Gordon et al., 2017). The importance of new particle formation and condensation from organic vapours to the global aerosol

load, cloud formation and climate have been getting increasing attention over that past 10 years (Glasius and Goldstein, 2016). However, there are still large uncertainties associated with these processes and this contributes to the overall uncertainty of aerosol particles' impact on climate (Kulmala et al., 2013).

In this paper, we investigate the potential climate feedback associated with increasing BVOC emissions due to increasing $CO_2$ concentrations and temperature, shown in Fig. 1. Note that the word feedback is used somewhat differently in this paper

compared to traditional climate science, since not only temperature, but also the $CO_2$ concentration is directly involved in the change in BVOC emissions. The increase in atmospheric $CO_2$ results in increasing temperature but also, gross primary productivity (GPP) through $CO_2$ fertilisation (Morison and Lawlor, 1999). Higher GPP results in more vegetation that can produce BVOC (Guenther et al., 1995). Increasing temperature also has a positive effect on the emissions of BVOC because of the exponential relationship between BVOC volatility and temperature (Kulmala et al., 2013). Additionally, rising levels of

$CO_2$ may have a direct impact on the BVOC emissions, as isoprene emissions have been found to decrease with increasing $CO_2$ levels (e.g. Wilkinson et al., 2009), but whether the same is true for monoterpenes is not yet clear (Arneth et al., 2016). Higher concentrations of BVOC gives an increase in aerosol number concentration ($N_a$) since oxidation products of BVOC contribute to new particle formation and early particle growth, as well as more secondary organic aerosols (SOA) mass due to increased condensation. The feedback loop then divides into two different branches.





The upper branch of the feedback loop involves aerosol effects on clouds, radiation and on temperature (the T-branch). The increase in SOA contributes to more cloud condensation nuclei (CCN), both through the formation of more aerosol particles, and through increased condensation, which increases the diameter of existing particles and makes them large enough to act as

seeds for cloud droplets (Kulmala et al., 2004). The increase in CCN will result in clouds with a higher cloud droplet number concentration (CDNC) and smaller droplets leading to a higher cloud albedo (Twomey, 1974). Smaller cloud droplets can also lead a delay in the onset of precipitation, which leads to a longer cloud lifetime (Albrecht, 1989). Higher cloud albedo and longer cloud lifetime leads to decreasing temperature, giving rise to a negative climate feedback.

The lower branch of the feedback involves the impact of aerosol particle scattering on GPP (The GPP-branch). More particles

and more aerosol mass means more scattering by aerosol particles in the atmosphere, which increases the fraction of diffuse radiation to global radiation (R). Increased fraction of diffuse radiation, at relative stable levels of total radiation, has been found to boosts photosynthesis through increased photosynthetically active radiation in shaded regions (Roderick et al., 2001). More photosynthesis increases the GPP, which result in larger emissions of BVOC and a positive feedback on BVOC emissions. Increased BVOC emissions have also been proposed to have other indirect forcing effects, e.g. on methane lifetime and ozone

concentrations, but these effects will not be investigated in this study.

Both measurement and modelling studies have previously investigated parts of the BVOC feedback shown in Fig. 1. Using long-term data of aerosol properties from eleven measurement stations, Paasonen et al. (2013) estimated the feedback associated with the T-loop, to globally be about -0.01 W m$^2$ K$^{-1}$. Scott et al. (2018) found a similar number (-0.013 W m$^2$ K$^{-1}$) using a global aerosol model together with an offline radiative transfer model. In Kulmala et al. (2014), the T-branch of the feedback was

estimated with an atmospheric model by doubling monoterpene emissions. This resulted in a global cloud radiative forcing of approximately -0.2 W m$^2$. Makkonen et al. (2012) found this number to be -0.5 W m$^2$ at lower anthropogenic aerosol emissions, using emissions from 2100 according to RCP4.5. The GPP-branch has been investigated using measurement data from a station in central Finland, which supported a statistically significant correlation between an increase in diffuse radiation ratio and higher aerosol loading during cloud free conditions, as well as a resulting increase in GPP (Kulmala et al., 2013,

2014). Rap et al. (2018) combined a global aerosol model, a radiation model and a land surface scheme and found the GPP-branch to contribute with a global BVOC emissions gain by 1.07. To our knowledge, no study has so far used an Earth System Model to investigate both branches of the BVOC-feedback.

This study provides a comprehensive global investigation of the BVOC-feedback using an Earth System Model. The model setup enables the vegetation and emissions in the land model to respond to changes in climate, $CO_2$ and radiation, capturing

diurnal as well as seasonal variations in the emissions of BVOC. Both emissions of isoprene and monoterpenes are calculated interactively by the land model and are included in the SOA formation in the atmospheric model. The scientific objectives of the study are to investigate the impact of $CO_2$ and temperature on the BVOC-feedback separately and combined. We aim to determine the importance of each step along the BVOC-feedback loop globally and regionally. Moreover, we want to determine the relative importance of the two branches of the feedback loop, as well as the overall relevance of the BVOC-feedback loop for estimating the future climate.



## 2 Method

### 2.1 Model Description

In this study, the Norwegian Earth System Model (NorESM) (Bentsen et al., 2013; Kirkevåg et al., 2013; Iversen et al., 2013)
has been used to investigate the feedback loop described in the previous section. NorESM is based on the Community Earth
System Model (CESM) but uses a different ocean model and a different aerosol module in the community atmosphere model
(CAM). The atmospheric model in NorESM is therefore called CAM-Oslo (Kirkevåg et al., 2013). We used CAM5.3-Oslo
(Kirkevåg et al., 2018) coupled to the Community Land Model version 4.5 (CLM4.5) (Oleson et al., 2013). CLM4.5 was run in
the BGC (biogeochemistry) mode, which includes active carbon and nitrogen biogeochemical cycling. In this mode, the plants
respond to changes in environmental conditions by enhanced or reduced growth, but the geographical vegetation distribution
does not change. Included in CLM4.5 is the Model of Emissions of Gases and Aerosols from Nature (MEGAN) version 2.1
model (Guenther et al., 2012) that provides emissions of BVOC from the plant functional types in CLM4.5. The BVOC in-
clude isoprene and the following compounds which are lumped together as monoterpenes in CAM-Oslo; myrcene, sabinene,
limonene, 3-carene, t-B-ocimene, $\beta$-pinene, $\alpha$-pinene. Both the vegetation and the emissions respond to changes in diffuse
radiation, $CO_2$ and other climate variables. $CO_2$-inhibition is included in MEGAN for isoprene (Guenther et al., 2012).

The aerosol scheme in CAM5.3-Oslo is called OsloAero (Kirkevåg et al., 2018) and has been developed at the Meteorological
Institute of Norway and the University of Oslo. OsloAero can be described as a "production tagged" aerosol scheme where
the aerosol tracers are defined according to their formation mechanism. The tracers includes 15 lognormal background modes,
which are modified by condensation, coagulation and cloud processing. CAM5.3-Oslo also includes some changes to the gas-
phase chemistry compared to CAM5.3. In CAM5.3-Oslo, isoprene and monoterpene can react with $O_3$, OH and $NO_3$. The
reaction between monoterpene and $O_3$ yields low volatile SOA (LVSOA) while the other five reactions between BVOC and the
oxidants yield semi-volatile SOA (SVSOA). The yields for the isoprene reactions are 0.05 and the yield for the monoterpene
reactions are 0.15, which reflects the findings in e.g. Jokinen et al. (2015). LVSOA and SVSOA can also be formed from
dimethyl sulphide as a proxy for methane sulphonic acid (MSA). Only the LVSOA takes part in the nucleation in the model
while the SVSOA condenses onto already formed aerosol particles (Makkonen et al., 2014). In NorESM, both LVSOA and
SVSOA are treated as non-volatile with condensation being kinetically limited.

The nucleation scheme was introduced into CAM-Oslo in Makkonen et al. (2014) but has since then been further developed
(Kirkevåg et al., 2018). The nucleation scheme includes binary homogeneous sulphuric acid-water nucleation (Vehkamäki
et al., 2002) as well as an activation type nucleation in the boundary layer. The activation type nucleation rate is calcu-
lated from the concentrations of $H_2SO_4$ and LVSOA available for nucleation according to eq. 18 ($J = 6.1 \times 10^{-7}[H_2SO_4] +$
$0.39 \times 10^{-7}[LVSOA]$) from Paasonen et al. (2010). The subsequent growth and survival to the smallest mode (median radius
23.6 nm) is modelled by a parameterization from (Lehtinen et al., 2007), depending mainly on the ratio between coagulation
sink and growth rate (from LVSOA and $H_2SO_4$). The treatment of early growth of aerosols has been adjusted in this version of
the model due to too high concentrations of particles from new particle formation. This was due to the survival percentage from
nucleation (radius 2 nm) to the smallest mode being unrealistically high. In OsloAero, coagulation is calculated only between



small modes and larger modes, while auto-coagulation and coagulation between smaller modes is considered negligible. In or-
der to improve this, we added coagulation onto all pre-existing particles to the coagulation sink used in the survival calculation
(Lehtinen et al., 2007).

The cloud schemes in CAM5.3-Oslo includes a deep-convection scheme (Zhang and McFarlane, 1995), a shallow convec-
tion scheme (Park and Bretherton, 2009) and the microphysical two moment scheme MG1.5 (Morrison and Gettelman, 2008;
Gettelman and Morrison, 2015) for stratiform clouds. The microphysical scheme includes aerosol activation according to
Abdul-Razzak and Ghan (2000) which depend on updraft velocity and the properties of the different aerosol modes. For both
liquid and ice, the mass and number are prognostic and the autoconversion scheme (Khairoutdinov and Kogan, 2000) includes
sub-grid variability of cloud water (Morrison and Gettelman, 2008).

## 2.2   Experimental Setup

In order to investigate the feedback-loop presented above, three different sets of experiments were performed with NorESM.
The first experiment was set up to simulate impacts of the change in BVOC emissions when plants respond to enhanced $CO_2$
concentrations. The $CO_2$ was doubled w.r.t year 2000 level (denoted $2xCO_2$), but note that the fixed SSTs highly restricted the
temperature increase from the radiative forcing associated with doubling the $CO_2$. The second experiment simulate the impact
of a warmer climate driven by a change in the sea surface temperature (SST) and sea ice to the year 2080 conditions according
to the RCP8.5 scenario (denoted $+\Delta SST$), but with fixed $CO_2$ concentrations at the year 2000. 2080 was chosen because the
$CO_2$ levels at this time is approximately equal to the $2xCO_2$ experiment. In the last experiment we doubled both the $CO_2$, and
changed the SSTs and sea ice as described previously ($2xCO_2+\Delta SST$). The experiments enable us to investigate the response of
the BVOC-feedback to increased $CO_2$ and temperature separately and then to see their combined effect in the last experiment.
Because the aerosol loading is expected to decrease in the future (Smith et al., 2016), we also ran a simulation identical to the
$2xCO_2+\Delta SST$, but where we changed the emissions of aerosol and precursor gases to PI levels, denoted $2xCO_2+\Delta SST$ LA
(low aerosol). This simulation was done in order to investigate whether the importance of the BVOC feedback will be larger if
the aerosol loading is smaller in the future. The doubling of $CO_2$, the SST increase and the reduction in aerosol emissions are
all at the top end of possible future scenarios and should thus not be viewed as the most likely future.

To be able to determine the importance of each step along the BVOC-feedback loop, each of the experiments described were
run with the feedback-loop turned on (FB-ON) and turned off (FB-OFF). In the FB-OFF simulations, we did not want changes
in $CO_2$, temperature or GPP to affect the BVOC emissions, essentially keeping concentrations constant at PD levels. This
was done by generating emissions fields from a control simulation and using these as input into the FB-OFF simulations, see
Fig. 2 and Table 1. We found that reproducing the diurnal variations in the BVOC emissions in the FB-OFF simulations was
important in order to get the BVOC concentrations in the model representative of those in the control simulation. The column
burdens of isoprene and monoterpene became much higher when no diurnal variation in the BVOC emissions was included
since the BVOC emissions were high also when the oxidant concentrations were low. Moreover, the reaction rates between the
BVOC and the oxidants are temperature dependent and thus lower during the nights. In order to produce emissions fields for
the FB-OFF simulations with correct diurnal variations, 6 years of control run emission data at half an hour time resolution





was averaged to create a yearly input file with half an hour time resolution (the time step used in the model). Thus, the FB-ON simulations and the FB-OFF simulations are set up the exactly the same except that the FB-ON simulations are run with
interactive BVOC emissions, while in the FB-OFF simulations the BVOC emissions are fixed at present day (PD) conditions, see Table 1.

Furthermore, to not have changes in weather patterns between the FB-ON and FB-OFF simulations mask the effects of the different BVOC emissions, we have used nudging (Kooperman et al., 2012) of horizontal winds and surface pressure (Zhang et al., 2014). Since meteorological conditions change significantly with doubling of $CO_2$ and temperature increase, the FB-
ON/FB-OFF simulations for each experiment are nudged to separate NorESM runs with the corresponding temperature/$CO_2$ changes (see Fig. 2 and Table 1). The nudging changes some of the meteorological variables in the model slightly, and therefore, also the control simulation (CTRL) from which the fixed BVOC emissions fields are generated was nudged to another CTRL simulation (see Fig. 2).

NorESM was run with a 1.9x2.5° horizontal resolution, 30 vertical levels and fixed sea ice and SSTs. The emissions of aerosols
and precursor gases were set to the year 2000 except for the simulations where we decrease the aerosol loading to PI levels, where the emissions from 1850 are used. Prescribed oxidant fields and land use at PD conditions are used for all simulations. The control simulation (CTRL) as well as the other four experiments described above were run for 30 years as a spin-up (see Fig. 2). After this, another 8 years were run to create the meteorological data for nudging for each experiment. The FB-ON simulations were initialised from the spin-up simulations and run for 8 years using nudging with a relaxation time of 6 hours.
The FB-OFF simulations were run in the same manner except that the BVOC emissions were read from file (as described above). The first two years of the FB-ON and FB-OFF simulations are considered a spin-up, due to the nudging and the change in the emissions in the FB-OFF simulations. Thus, the last 6 years of the simulations are used for the analysis.

## 3  Results and Discussions

### 3.1  BVOC emissions and SOA

We will start by discussing the part of the BVOC feedback common to the two branches and then discuss each branch of the feedback separately.

#### 3.1.1  BVOC Emissions

The BVOC emissions calculated by NorESM are in line with previous studies. In the CTRL run, the BVOC emissions are 366 Tg yr$^{-1}$ for isoprene and 115 Tg yr$^{-1}$ for monoterpenes. These values are in the range of those in Guenther et al. (2012) for
monoterpenes, but in the lower end for isoprene. For the 2x$CO_2$+$\Delta$SST FB-ON simulation the emissions are 586 Tg yr$^{-1}$ (+60 %) for isoprene and 198 Tg yr$^{-1}$ (+73 %) for monoterpenes. The emissions are somewhat lower than estimated for the future climate in previous studies (Laothawornkitkul et al., 2009) but the relative increases are in the high end (Carslaw et al., 2010). The isoprene emissions increase more when the temperature is increased (+$\Delta$SST) than when the $CO_2$ is doubled, but





the opposite is true for monoterpenes, see Fig. 3 c and d.

The emissions of isoprene and monoterpenes are higher almost everywhere in the FB-ON simulations with $2xCO_2$, $+\Delta SST$ and $2xCO_2 + \Delta SST$ than in the FB-OFF simulations with the same setup (see Fig. 3 and S1). The absolute increase in the emissions is largest over the tropical forests while the relative increase in emissions is greatest over the boreal forests in the Northern Hemisphere (NH). Generally, the $CO_2$-inhibition of isoprene is masked by the $CO_2$ and temperature boosts of the vegetation, which leads to a higher leaf area index (LAI) and GPP. In the experiment with only increased $CO_2$, there are a few areas in Africa and India that seem to have lower isoprene emissions due to $CO_2$-inhibition. This can be seen as lower isoprene emissions and higher monoterpene emissions in the same place, Fig. S1 (a and c). This does not occur in the experiments where also the SST are increased. Over some regions in the tropics (parts of the Africa and the Amazon), especially in the $+\Delta SST$ experiment, both monoterpene and isoprene emissions decrease. This is caused by a decrease in the LAI associated with plant mortality that seem to occur because of heat stress. The decrease in LAI leads to a lower albedo in these forest regions, which further increases the temperature, causing more heat stress and creating a feedback mechanism on the vegetation. Nevertheless, the vegetation has had time to adapt to the new temperatures and stabilise by the end of the 30-year spin-up period. The decreases in LAI is smaller in the $2xCO_2 + \Delta SST$ experiments as the vegetation is seeded by $CO_2$ (Fig. 3a).

### 3.1.2 SOA

The higher BVOC emissions in the FB-ON simulations lead to larger SOA production (see Fig. 4b). The SOA production in the CTRL simulation is 75 Tg yr$^{-1}$ which is in the range of previously estimated by global models (Tsigaridis et al., 2014; Glasius and Goldstein, 2016). The SOA production in the FB-ON simulations is similar for the $2xCO_2$ and $+\Delta SST$ experiments (90 and 92 Tg yr$^{-1}$) while the combined effect of higher $CO_2$ and temperature gives a higher SOA production, with values of 115 Tg yr$^{-1}$. The column burden of SOA is higher over the entire globe when the BVOC-feedback is on compared to when it is turned off, except in the $+\Delta SST$ experiment over and downwind of the regions where the BVOC emissions decrease, see Fig. 4 a and S2 b. The largest absolute increase column burden SOA is over the tropical forests while the largest relative increases are over the Arctic and sub-Arctic. The fraction of SOA in the aerosol particles is also higher when the feedback is turned on which leads to a reduction in the hygroscopicity of the particles (not shown).

### 3.1.3 Aerosol number and Size

Not only the mass of the aerosol particles is affected by higher levels of BVOC, but also the number concentration of aerosol particles and their sizes. The changes in the number concentration and size of the particles vary with region. The largest difference in $N_a$ between the FB-ON and FB-OFF simulations occurs over, and downwind of, the tropical rain forests, as well as over the boreal forests in the NH (see Fig. 4 c). The relative difference is largest over the boreal forests in the NH where the particle number concentrations are generally low. The largest absolute differences on the other hand occur in the Tropics. Over regions where the emissions decrease (in the $+\Delta SST$ experiment), the $N_a$ decreases (Fig. S2 d).

In order to investigate the effect on the sizes of the particles, we analysed the averaged boundary layer aerosol size distributions for two of the regions most affected by the feedback, the boreal forests and the tropical islands in South East Asia. The size





distributions are created from the number median radius and standard deviations of the 12 particle mixtures in OsloAero (Kirkevåg et al., 2018). Over the boreal forests, the higher BVOC emissions result in more particles in the Aitken mode Fig. 5 a and b. The enhanced growth of the particles also result in more particles in the accumulation mode and in a shift to larger sizes of the Aitken mode, which result in a small decrease in the number of particles below 25 nm. In the Tropics, there is a larger (smaller) absolute (relative) increase in Aitken mode particles. The shift in the size distribution due to more condensing vapours is larger here than over the boreal forests and results in decreasing particle concentrations up to 70 nm. The biggest changes in both number and shift in size distribution is seen in the $2xCO_2+\Delta SST$ experiment. The geographical differences in the particle sizes are located further away from the sources than the differences in number concentrations, in particular in the Tropics (not shown).

### 3.2 The T-feedback branch

#### 3.2.1 CCN

The CCN response of the feedback is a combination of the changes in $N_a$, particle sizes and hygroscopicity. The CCN concentrations are generally higher when the feedback is turned on. However, at low supersaturations (0.2%), the CCN concentration over some regions (in particular over the boreal forests), is lower in the simulations with the feedback turned on (Fig. 6a). The cause for this is the large amount of Aitken mode particles formed through new particle formation. The smaller particles compete with the larger particles for the water vapour, which reduces the amount of aerosol particles that can activate into cloud droplets at low supersaturations. The concentrations of CCN in these regions are very low and the absolute decrease in CCN is small. Moreover, it should be noted that the CCN concentration in the model is calculated only for the cloud free areas in the grid boxes. Thus, the particles that are activated into cloud droplets are not included in the CCN concentrations. At higher supersaturations, (1 %), also particles at smaller sizes can be activated and thus the feedback results in more CCN almost everywhere (Fig. 6c). The areas downwind of the tropics, where the feedback mainly results in an increase in particle size, have higher CCN at both levels of supersaturation. The effect of increasing particle sizes and number generally dominate the effect of decreased particle hygroscopicity since the feedback contribute with increasing number of CCN.

#### 3.2.2 Cloud properties

The effect from the BVOC feedback on the clouds are mainly seen over and downwind of the regions where the BVOC emissions change the most. The vertically averaged CDNC generally increase, mainly north of 45° N and in the Tropics Fig. 7a. The weakest response of the CDNC to the feedback occur in the experiment where only $CO_2$ has been changed (Fig. 7b and S4). In the experiment with only increased SST, the CDNC is higher mainly in the northern hemisphere since the BVOC emissions in parts of the Tropics decrease (Fig. S4 b). The higher levels of CDNC occur predominantly during the local summer when the BVOC emissions are the highest.

The increasing CDNC associated with the feedback is accompanied by a decrease in cloud droplet effective radius ($r_e$) and an increasing cloud water path (CWP) (Fig. 7 c and e). The total cloud fraction does however not seem to be impacted to the same





extent, see Fig. 7 g and h, which may be an effect of the nudging. There is an increase in the CF over the boreal forests, mainly during winter, by up to 4 %. In summer, there is an increase in low and mid-level clouds over the Arctic and NH mid-latitudes.

This is accompanied by a decrease in the high-level clouds and does therefore not show up clearly in the Fig. 7 g. In the Tropics, there are no systematic changes in the cloud fraction as a result of the feedback.

The strongest and most widespread difference in the cloud microphysical effects occur at the NH mid- and high latitudes. One cause for this is the cloud cover and cloud types present close to the emission regions. The clouds in the mid- and high latitudes are commonly stratiform, for which the model includes $N_a$ in the calculations of CDNC (through the Abdul-Razzak

and Ghan (2000) scheme for activation). The differences in CDNC are not as widespread in the Tropics, since shallow and deep convection (which aerosols generally do not affect in ESMs) are the dominant cloud types here. Another cause for the more widespread cloud changes in the NH is the larger land areas here, i.e. larger areas where the emissions differ.

### 3.2.3 Cloud forcing

The potential of the BVOC feedback to affect future climate will now be evaluated by investigating the changes in cloud forcing

between the FB-ON and FB-OFF simulations. Since we cannot determine the full temperature response of the feedback, the differences in forcing between the FB-ON and FB-OFF simulations will be used to estimate the potential climate impact of the changed cloud properties. The patterns of the difference in the cloud forcing between the simulations with the FB turned on and the FB turned off (Fig. 8 a and c) resemble the patters of the difference in CDNC (Fig. 7 a). The higher CDNC in the high and mid latitudes, associated with the FB is accompanied by a decrease in the $NCF_{Ghan}$ (Net Cloud Forcing calculated

using the method by Ghan (2013)) by up to -11 W m$^{-2}$, see Fig. 8a. The effect of the feedback is seen mainly during the local summer when the BVOC emissions are the highest. The differences in $NCF_{Ghan}$ are smallest in the 2xCO$_2$ experiment and strongest in the 2xCO$_2$+$\Delta$SST experiment (Fig. 8 and S5.)

The feedback does not only contribute with an enhanced negative cloud forcing though. The difference in NCF at the surface (NCFS) between the FB-ON and FB-OFF simulations is positive over the NH boreal forests during winter, in the experiments

with increased SST (Fig. 8 e and S5 f). The changes in microphysical properties as well as cloud cover leads to an increase in the positive long wave cloud forcing (LWCF) at the surface, which is larger than the corresponding increase in negative short wave cloud forcing (SWCF). It can be concluded that the BVOC feedback can contribute to both enhanced and reduced negative cloud forcing depending on region and season. Nevertheless, the yearly global average $NCF_{Ghan}$ is -0.43 W m$^{-2}$ ($SWCF_{Ghan}$ -0.45 W m$^{-2}$, $LWCF_{Ghan}$ 0.02 W m$^{-2}$) when the FB is turned on in the 2xCO$_2$+$\Delta$SST experiments, indicating

that the feedback can contribute with a potentially important impact on the future climate on a global scale.

The strongest and most widespread negative cloud forcing associated with the feedback is seen in the Arctic during summer. This is interesting since the Arctic is currently, and is expected to continue experiencing, the largest warming in response to the increasing atmospheric concentrations of greenhouse gases (IPCC, 2013). The strong impact of the BVOC feedback in the Arctic during summer could possibly counteract part of this Arctic amplification. The large impact of the feedback in the NH mid and high latitudes also results in a quite large difference in the effect of the feedback between the hemispheres. The



difference in the $NCF_{Ghan}$, between the FB-ON and FB-OFF simulations for the 2x$CO_2$+$\Delta$SST experiments, is -0.56 W m$^{-2}$ in the NH, while in the SH it is -0.30 W m$^{-2}$.

## 3.3 The GPP-feedback branch

### 3.3.1 AOD

The higher aerosol loading associated with the feedback also result in higher values for the aerosol optical depth (AOD). The largest relative differences between the FB-ON and FB-OFF simulations occur over, and downwind, the tropical forest and the boreal forests in in the NH, see Fig. 9 a. The AOD effects are largest in the local summer when the emissions are the highest.

### 3.3.2 Diffuse radiation

The ratio between the diffuse radiation and the global radiation is, according the BVOC-feedback hypotheses, expected to increase with higher aerosol scattering. Our model simulations show only a small relative difference in R (maximum 5 %) between the FB-ON and FB-OFF simulations (Fig. 9 c). The regions where there is a strong difference in R between the FB-ON and FB-OFF simulation corresponds to the regions with the largest change in AOD. However, a statistical analyses of the differences between the monthly means from the FB-ON and FB-OFF simulations show that the correlation coefficient between the difference in R and the difference in total cloud cover (0.53) is much higher than between the difference in R and the difference in AOD (0.08), see Fig. 10 a and b. Small changes in the cloud cover can offset the AOD effects on R. Changes in cloud cover can therefore explain the decreases in R over e.g. Scandinavia (Fig. 9) even though the AOD increases there.

### 3.3.3 GPP

Next, we will investigate the relationship between R and GPP. Neither in the maps nor in the statistical analyses do we find any strong relationship between R and GPP, see Fig. 9 e and 10 c. The positive effect of diffuse radiation on vegetation growth is included in CLM (Oleson et al., 2013) but it seems like other factors perturbed by the T-branch are affecting the vegetation more. Moreover, the difference in R between the FB-ON and FB-OFF simulations was quite small. The relationship between R and GPP is also affected by changes in the total amount of radiation. If the total radiation decreases sufficiently, an increase in R will not boost GPP (Knohl and Baldocchi, 2008). There is a negative correlation between the change in R and the change in the total visible radiation in our experiments. The hypothesised boost of GPP by R might therefore be masked by the change in the total visible radiation. Since the focus of this study is the effect of the feedback on a global scale, we have chosen not to look into if we can find the effect of R on GPP in certain conditions or locations.

The GPP instead seem to respond to changes associated with the T-feedback branch (Fig. 10 d). In particular, there is a decrease of GPP in the sub-Arctic during the summer months associated with lower temperatures caused by the enhanced negative $NCF_{Ghan}$. Even though we are running with fixed SSTs, the temperatures over land can change somewhat in response to the changed forcing. In addition, a decrease in total visible radiation reaching the vegetation, associated with the increase in low-cloud cover in this region, can contribute to the decrease in GPP. Overall, the GPP is slightly lower in the simulations where





we include the feedback. These results are in contrast to the results by Rap et al. (2018), which did not include the effects from the T-branch in their study. In our study, it seems that the effects from the T-branch of the BVOC-feedback loop is dominating over the GPP-branch. The GPP-branch may however be important on local scales not resolvable by NorESM.

## 3.4 Direct Aerosol forcing

The scattering of radiation from aerosols in the atmosphere did not seem impact the GPP significantly in our experiments, but we do find a direct impact on climate. The annual average net direct aerosol forcing calculated using the Ghan (2013) method ($NDF_{Ghan}$) is locally down to -2.2 W m$^{-2}$ when the feedback is turned on, see Fig. 11 b and c. The largest differences in $NDF_{Ghan}$ between the FB-ON and FB-OFF simulations is seen close to the sources and over the regions that have large absolute changes in the emissions, i.e. the Tropics. Globally averaged, the difference in NDF is -0.06 W m$^{-2}$ for the 2xCO$_2$+$\Delta$SST experiment. This is approximately 15 % of the difference in forcing from the clouds. The magnitude of the differences in the $NDF_{Ghan}$ indicate that the BVOC feedback can provide an, at least regionally, enhanced negative forcing also through the direct aerosol forcing.

## 3.5 Future lower aerosol loading

In order to investigate how the impact of the feedback changes if the aerosol emissions decrease in the future, we also ran the 2xCO$_2$+$\Delta$SST experiment with lower anthropogenic aerosol emissions. The BVOC emissions in 2xCO$_2$+$\Delta$SST LA FB-ON simulation are almost the same as in the 2xCO$_2$+$\Delta$SST FB-ON simulation (4 and 3 % higher for isoprene and monoterpenes). The response to the feedback is however larger in the experiment with lower anthropogenic emissions. The relative differences in N$_a$ is larger, especially over regions with large anthropogenic emissions in PD. This indicate that BVOC will be more important for aerosol formation in the future, if the anthropogenic emissions decrease. The relative CDNC difference is also greater in the experiment with low anthropogenic emissions in both the Tropics and the NH. There are areas (such as South East Asia) were the relative differences in CDNC is close to zero in the 2xCO$_2$+$\Delta$SST experiment and up to 30 % in the 2xCO$_2$+$\Delta$SST LA experiment. That the effects on the clouds are largest in the 2xCO$_2$+$\Delta$SST LA experiment is not surprising since clouds formed in clean condition are most susceptible to aerosol perturbations (Spracklen and Rap, 2013).

The stronger BVOC impact on the clouds in the experiment with lower aerosol loading result in a larger impact from the feedback on the radiation budget. The difference in the yearly global average $NCF_{Ghan}$ for the 2xCO$_2$+$\Delta$SST LA is 53 % higher than for the 2xCO$_2$+$\Delta$SST experiment, see Table 2. In addition, the direct effect associated with the feedback is larger when the anthropogenic aerosol load is reduced. The difference in $NDF_{Ghan}$ is 29 % higher for the experiment with lower aerosol loading. These results show that the importance of the BVOC feedback will become substantially more important if, as expected, the anthropogenic aerosol emissions are reduced in the future. These results are interesting, especially since some large emitters have already started reducing their SO$_2$ emissions (Li et al., 2017). The total aerosol forcing associated with the feedback in the 2xCO$_2$+$\Delta$SST (LA) experiment is -0.49 (-0.73) W m$^{-2}$ which is 13 (20) % part of the positive radiative forcing (calculated according to (Myhre et al., 1998)) associated with a similar doubling of CO$_2$.



### 3.6 Limitations and uncertainties

The investigation of the effects of BVOC is challenging since it involves complex interactions not only in the atmosphere, but also in the biosphere. In this investigation, the focus has been on the potential atmospheric consequences of increased BVOC emissions. However, the future BVOC emissions are highly sensitive to what will happen to the vegetation. This was clearly seen in our simulations where we increased only the SST and found that GPP is reduced in several regions due to heat stress. This cancels or even reverses the BVOC feedback in these regions. How future vegetation will respond to climate change is still highly uncertain (Friend et al., 2014).

Our simulations does not allow changes in the distribution of the vegetation and therefore does not include any effects of geographical shifts in vegetation. A poleward shift in the vegetation could increase the BVOC emissions in these regions (Peñuelas and Staudt, 2010). Nevertheless, changes in surface albedo, as well as latent and sensible heat fluxes associated with such shifts (Bonan, 2008) could counteract/dominate parts of the effects seen from the increased emissions BVOC. Changes in land use also has the potential to affect the BVOC emissions, but has not been taken into account in this study. A recent study by Hantson et al. (2017) including land use, found no increase in BVOC emissions at the end of the century. However, they also note that the land use scenarios are highly uncertain.

There are also uncertainties associated with the emissions from the plants themselves. In MEGAN2.1, used in this study, $CO_2$ inhibition is included for isoprene. There are indications that the inhibition also affect monoterpenes and some studies include it also for monoterpenes (Arneth et al., 2016). Including $CO_2$ inhibition for monoterpenes could have reduced the difference is monoterpene emissions between the FB-ON and FB-OFF simulations and reduced the effect of the feedback. Plant stress due to heat or insect infestations can affect the magnitude and type of BVOC emissions (Zhao et al., 2017). These effects are very complex and have not been included in this study.

During the setup of the experiments of this study, we found that the model was sensitive to the diurnal variation in the BVOC emissions (also described in Sect. 2.2). The column burden of isoprene (monoterpene) was, on a global average, 57 (13) % higher when monthly averaged emission files without diurnal variation was used in the model instead of using the interactive emissions. Adding a diurnal variation (the one included in CAM5.3) to the monthly emissions field improves the column burden values for isoprene, but for monoterpenes, the column burdens stay high. The resulting difference in the column burden of SOA (+5 % on a global average) is dampened by complex processes associated with nucleation and condensation. However, the lack of auto-correlation between the emissions and oxidants (when using monthly emissions) can results in longer lifetimes for the BVOC and a shift in region and level where the SOA formation occur. This has been shown to affect the indirect aerosol effect (Karset et al., 2018). Monthly BVOC emission files should therefore be used with caution. In this study, prescribed oxidant fields at PD conditions with applied diurnal variation for OH and $HO_2$ was used. Running the model with more advanced gas phase chemistry would have simulated the interactions between the BVOC and the oxidants more realistically.

New particle formation, BVOC and SOA parameterizations are now implemented in many ESMs but are still under development and associated with uncertainties (e.g., Tsigaridis and Kanakidou, 2018; Makkonen et al., 2014; Gordon et al., 2016). The BVOC feedback mechanism is highly sensitive to the parameterizations associated with new particle and SOA formation.



The yields associated with the formation of LVSOA and SVSOA from monoterpenes and isoprene are largely uncertain, which may significantly affect the feedback. The parameterizations of nucleation rates and early growth of the particles can also have a strong impact on the simulations of the feedback. Moreover, the SOA-scheme in NorESM does not account for effects of

temperature on partitioning of SOA-precursors. Warmer temperatures might lead to less SOA formation with same amount of precursors, which would reduce the feedback. In addition, the SOA formation from biogenic precursors could be highly susceptible to modification by anthropogenic emissions of VOC (Spracklen et al., 2011), which are not currently included in NorESM. We hope that the importance of the feedback found in this study will inspire further the development of these parameterizations in ESMs.

Running the model with fixed SSTs and nudging provides a nice setup to study each step in the feedback loops at low computational cost, but it also comes with some limitations. The nudging enabled us to run the FB-ON and FB-OFF simulations with the same meteorological conditions. We can therefore conclude that the difference between the simulations were only associated with the BVOC emissions and the feedback and not caused by natural variability. The nudging does however mean that any impacts of the feedback on horizontal winds and pressure is not captured in this investigation. Moreover, the fixed

SSTs and sea ice limit the temperature response to the feedback. There is some temperature response to forcing induced by the feedback over land, but not over the oceans. The second order feedbacks, such as decreasing BVOC emissions associated with the temperature decrease due to the enhanced negative cloud and direct forcing will not be properly simulated with this setup. Investigating the feedback with free running simulations using a coupled version of NorESM would be a very nice complement to this study.

In this paper, we have focused on the BVOC feedback mechanisms shown in Fig. 1, but there are other indirect effects of BVOC that could influence the feedback that are not included in this study. Two such effect involves impacts on ozone production and methane lifetime. When BVOCs are oxidized in the atmosphere, they affect the chemical composition as well as the oxidization capacity of the atmosphere. Firstly, VOCs can contribute to enhanced ozone production if sufficient $NO_x$ is available, while they can give a net consumption in low $NO_x$ conditions (Monks et al., 2015). Secondly, the oxidation of BVOCs can

decrease the oxidation capacity of the atmosphere, thus increasing the atmospheric lifetime of methane. Both of these effects could result in a positive radiative forcing with increased BVOC emissions. Moreover, some of the processes in the BVOC feedback investigated here may affect the carbon budget, however, such effects are out of the scope of this manuscript.

## 4  Conclusions

An ESM has been used to investigate two feedbacks induced by increased emissions of BVOC in response to higher $CO_2$

concentrations and/or temperature (Fig. 1). We find that higher BVOC emissions indeed lead to the formation of more SOA mass, as well as both higher aerosol number concentrations and larger particle sizes. This leads to clouds with more and smaller droplets and higher cloud water path. The changes in the clouds are found to contribute with an enhanced negative cloud forcing, confirming the possibility for BVOC to contribute with a negative climate feedback. The feedback is strongest over and downwind of the boreal and Tropical forests. Solely increasing the $CO_2$ levels produces a somewhat weaker feedback





response than solely increasing the temperatures, but the strongest response comes from increasing both $CO_2$ and temperature. In this investigation, we do not find that the enhanced aerosol scattering leads to a boost of GPP globally (see Fig. 1). The response of the GPP is instead dominated by the BVOC induced changes of the clouds. The enhanced aerosol scattering

associated with the feedback is however found to lead to a stronger negative forcing (direct effect). We would therefore suggest modifying the BVOC feedback in Fig. 1 as can be seen in Fig. 12. Because the GPP seem to be more affected by the cloud changes than the AOD changes, the arrows between AOD and GPP has been dashed. However, AOD can now be seen having a negative feedback on temperature. The combined effects from both altered cloud properties and AOD is found to contribute with a negative radiative effect of -0.47 W m$^{-2}$. To put this number in context, the radiative forcing from a doubling of $CO_2$

is about 3.7 Wm$^{-2}$. Thus, this feedback could offset this by about 13 %, or even up to 20 % given a strong reduction in anthropogenic aerosols. This leads us to conclude that the BVOC feedback is very relevant for estimating climate sensitivity with ESMs and to provide model-based projections of the future climate.

There are still large uncertainties associated with the processes associated with the BVOC feedback, both in models and measurements. The aim of this study was not to provide a final answer to the importance of the feedback. Instead, we wanted

to use the current knowledge implemented in NorESM to test the potential importance of including these processes in an ESM when predicting the future climate. The results from this study should encourage and inspire further research to improve the representation of these processes in ESM.

*Author contributions.* M.K.S. performed the model simulations, did the data analysis and wrote the manuscript. I.H.H.K. provided support during the setup of the model. S.M.B., I.H.K.K., R.M. and T.K.B. contributed with discussions regarding the experimental design, data

analysis and manuscript.

*Competing interests.* The authors declare that they have no conflict of interest.

*Acknowledgements.* The research leading to these results has received funding from the European Union's Seventh Framework Programme (FP7/2007-2013) project BACCHUS under grant agreement no 603445. This work was supported by LATICE, a strategic research area funded by the Faculty of Mathematics and Natural Sciences at the University of Oslo. I.H.H. Karset has been financed by the research council

of Norway (RCN) through the project EVA and the NOTUR/Norstore projects (Sigma2 account: nn2345k, Norstore account: NS2345K). We would like to thank Alf Kirkevåg and Øivind Sealand for support in the work with NorESM.



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





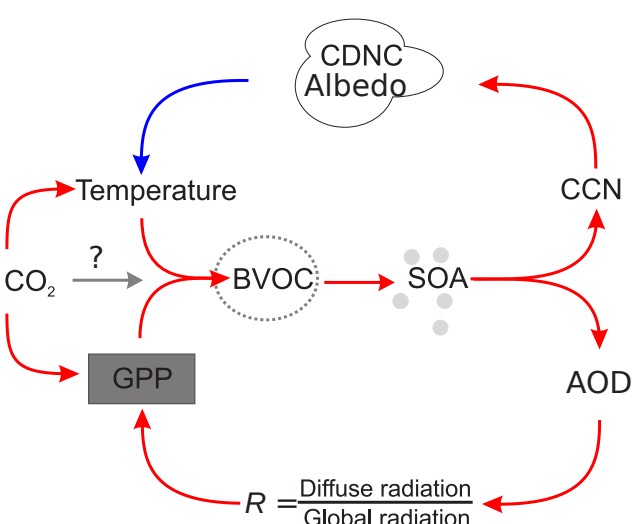

**Figure 1.** The BVOC feedback driven by increasing $CO_2$ and temperature. The upper branch if the feedback is the T-branch while the lower part is the GPP-branch. Modified after Kulmala et al. (2014)



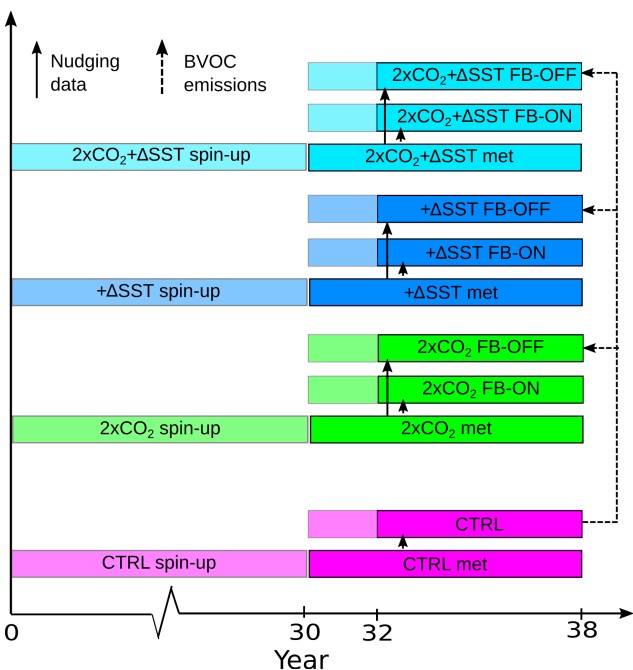

**Figure 2.** The simulation setup. The CTRL has $CO_2$ and SSTs at PD levels. The $2xCO_2$ simulations has doubled $CO_2$ w.r.t the year 2000. In the $+\Delta SST$ simulations the SST and sea ice are increased to the year 2080 levels. In the $2xCO_2+\Delta SST$ simulation the $CO_2$ is doubled and the SST and sea ice are changed to the year 2080 levels. The CTRL, as well as all FB-ON and FB-OFF simulations, are nudged to their respective met simulation. All FB-ON simulations has interactive emissions while the FB-OFF simulations has fixed emissions from the CTRL simulation.



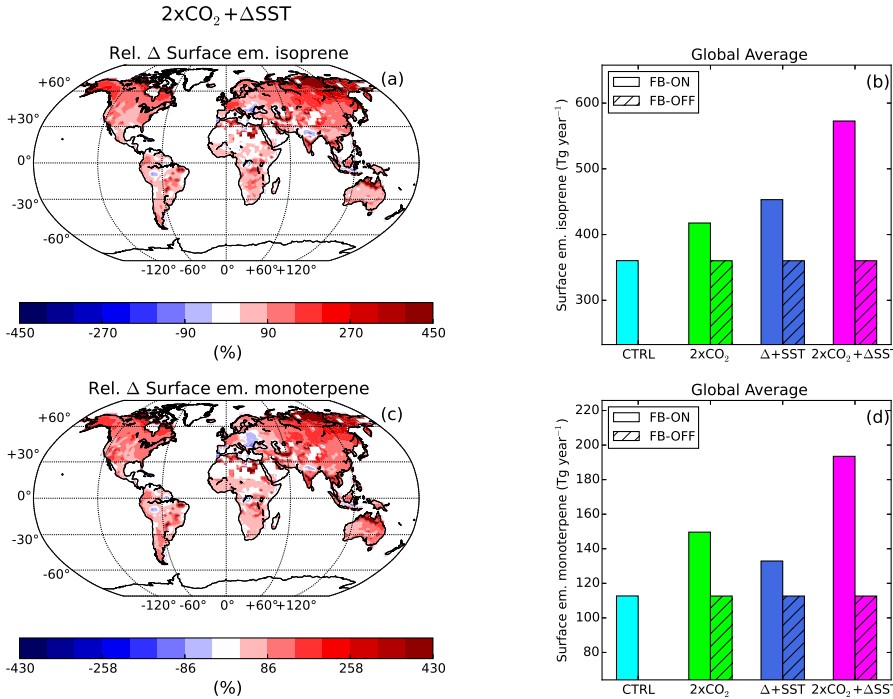

**Figure 3.** The relative difference between the FB-ON and FB-OFF simulations of the annual average surface emissions of isoprene (a) and monoterpenes (c) for the $2xCO_2+\Delta SST$ experiment. The relative difference is defined as the (FB-ON - FB-OFF) / FB-OFF. In the bar plots, the yearly global surface emissions of isoprene (b) and monoterpenes (d) for the CTRL simulation as well as the three experiments (both FB-ON and FB-OFF simulations) are shown.



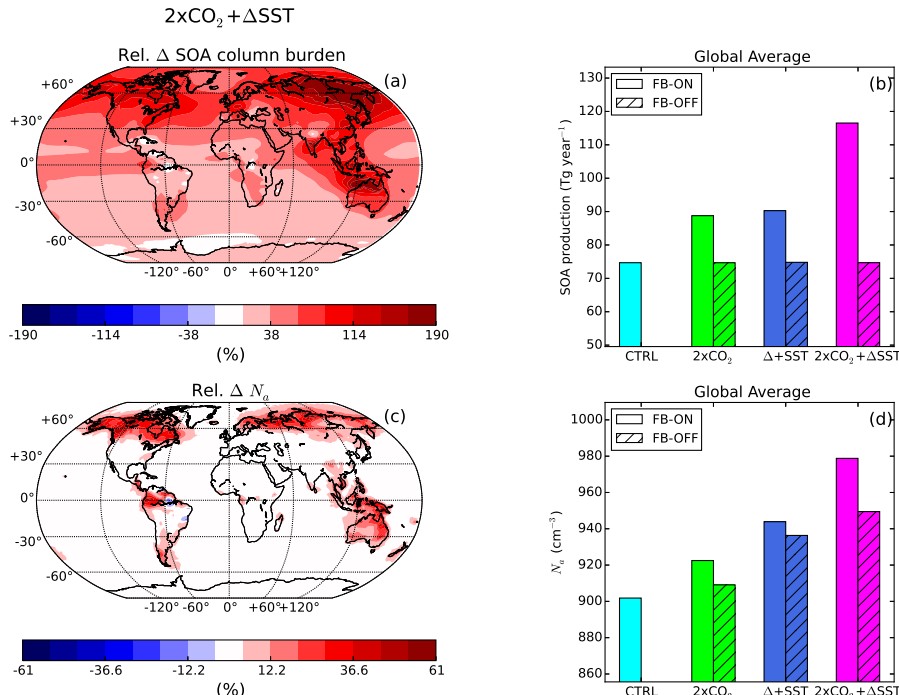

**Figure 4.** The relative difference between the FB-ON and FB-OFF simulations in the annual average column burden SOA (a) and $N_a$ in the boundary layer (c) for the 2xCO$_2$+$\Delta$SST experiment. In the bar plots, the average yearly global production of SOA (b) and the global average $N_a$ in the boundary layer (d) are shown, for the CTRL simulation as well as the three experiments (both FB-ON and FB-OFF simulations).





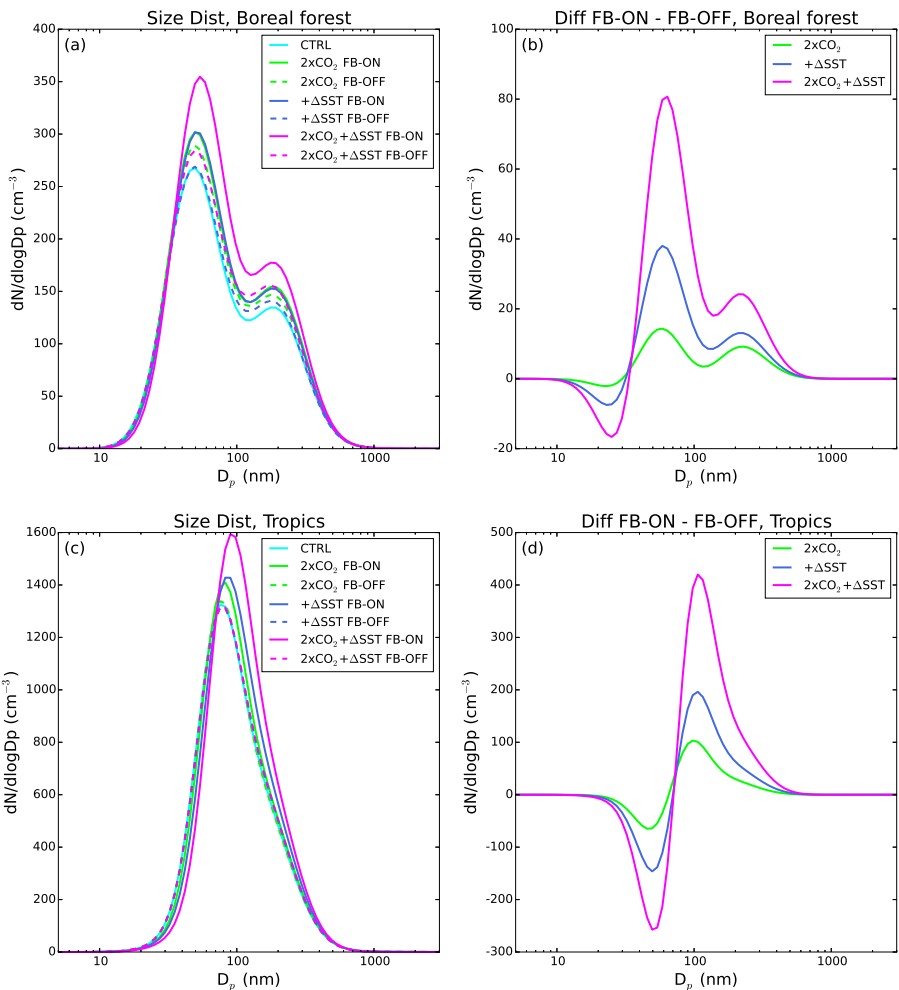

**Figure 5.** Annually averaged aerosol number size distributions in the boundary layer for the Boreal forest region (lat 55 to 70, lon: -180 to 180) and the region around the Tropical islands in South East Asia (lat: -20 to 20, lon: 90-130). In a and c, the distributions from the CTRL and the three experiments are plotted while in b and d, the difference between the FB-ON and FB-OFF simulations are plotted.



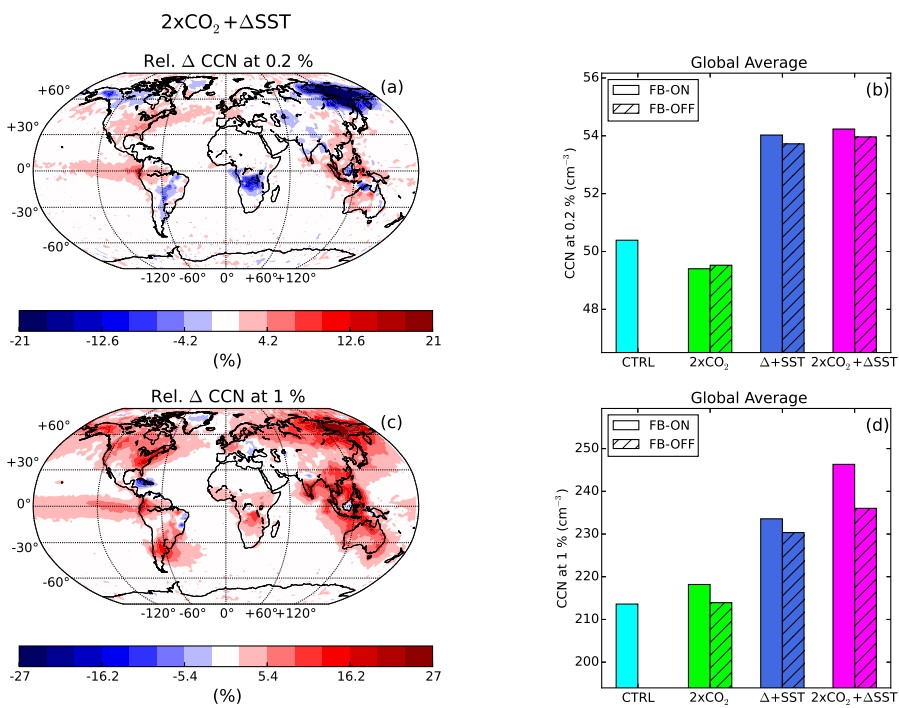

**Figure 6.** The relative difference between the FB-ON and FB-OFF simulations in the annual average CCN at 0.2 % (a) and 1 % (c) in the boundary layer, for the 2xCO₂+ΔSST experiment. In the bar plots, the globally averaged CCN at 0.2 % (b) and 1 % (d) in the boundary layer are shown, for the CTRL simulation as well as the three experiments (both FB-ON and FB-OFF simulations).





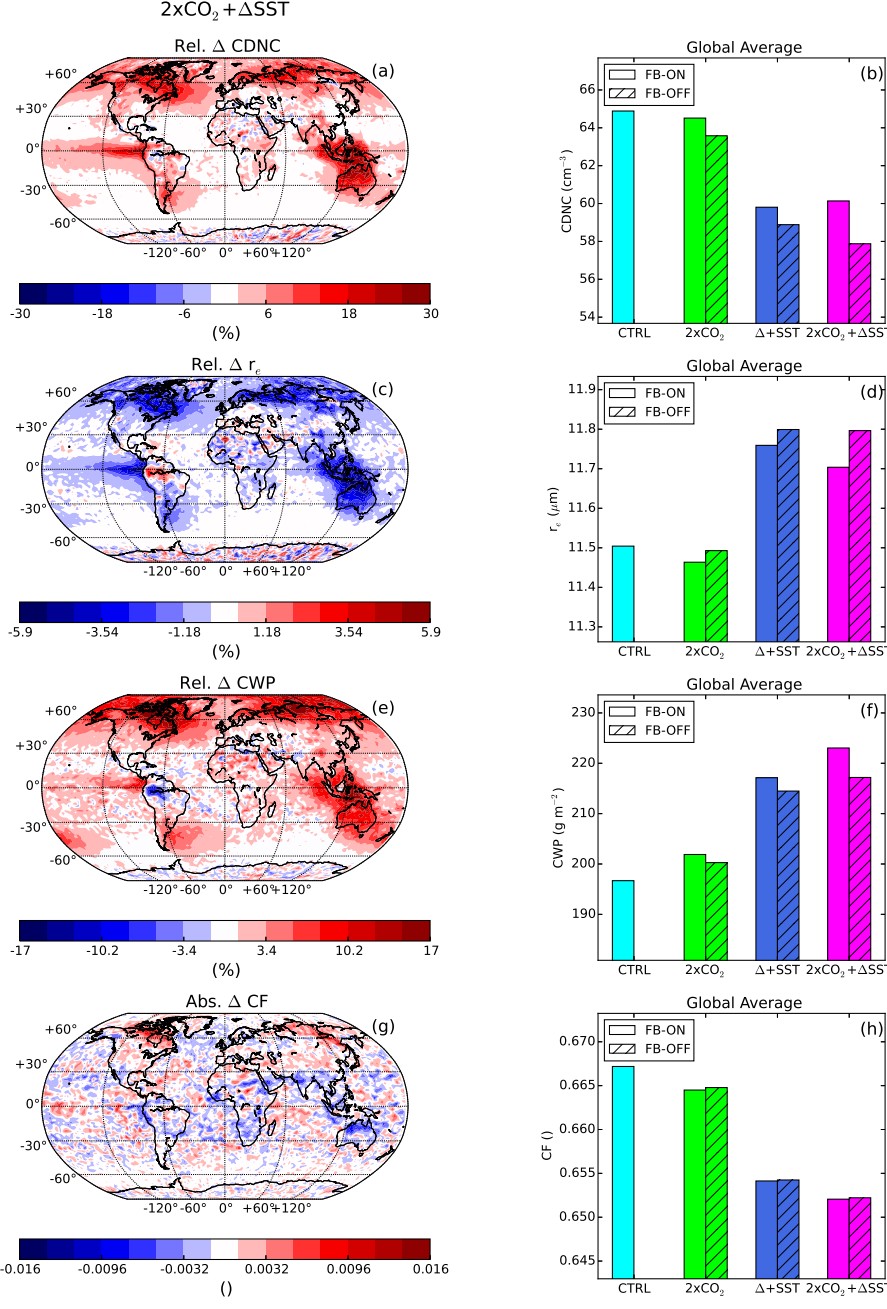

**Figure 7.** The relative/absolute difference between the FB-ON and FB-OFF simulations in the annual vertically averaged CDNC (a), the vertically averaged $r_e$ (c), the CWP (e) and the total CF (g) for the $2xCO_2 + \Delta SST$ experiment. In the bar plots (b, d, f, h) the globally averaged values of the same variables are shown, for the CTRL simulation as well as the three experiments (both FB-ON and FB-OFF simulations). For the CDNC, $r_e$ and CWP, the in-cloud values are used.





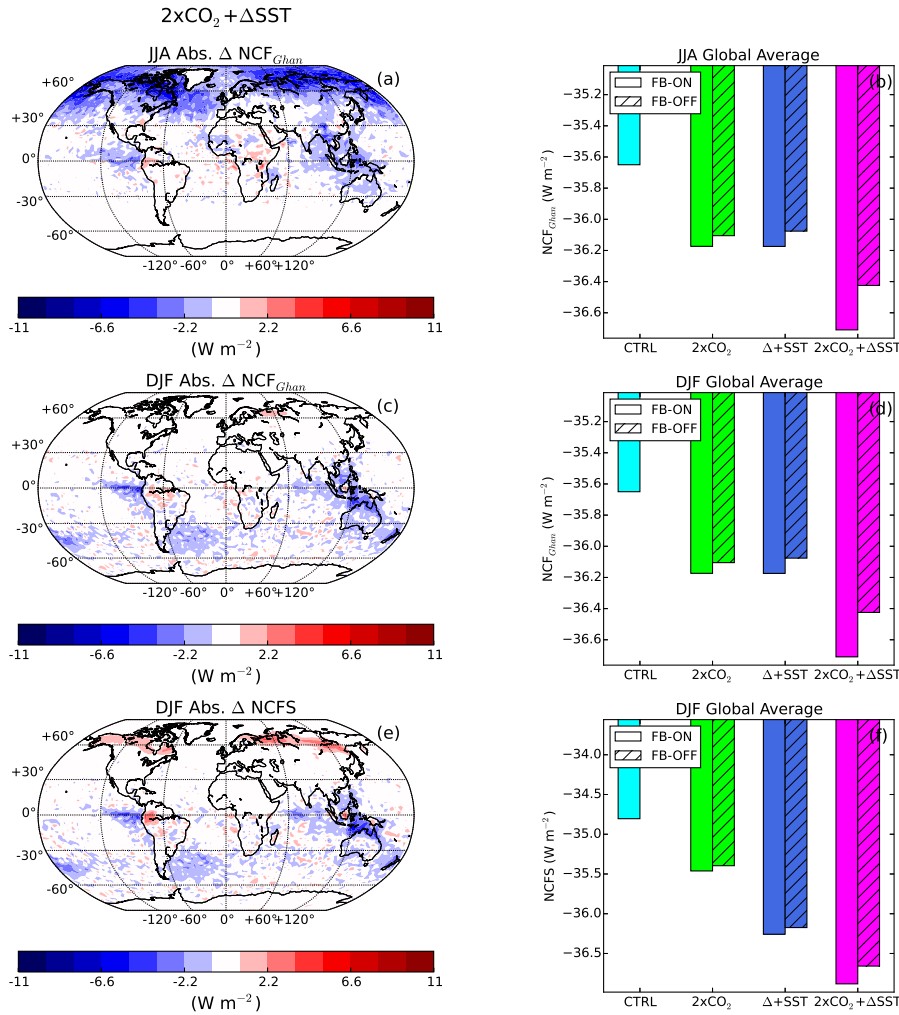

**Figure 8.** The absolute difference between the FB-ON and FB-OFF simulations for the $NCF_{Ghan}$ during June, July and August (a), December January and February (c) as well as the NCFS during December January and February (e) for the $2xCO_2+\Delta SST$ experiment. In the bar plots (b, d, f), the globally averaged values of the same variables are shown, for the CTRL simulation as well as the three experiments (both FB-ON and FB-OFF simulations).




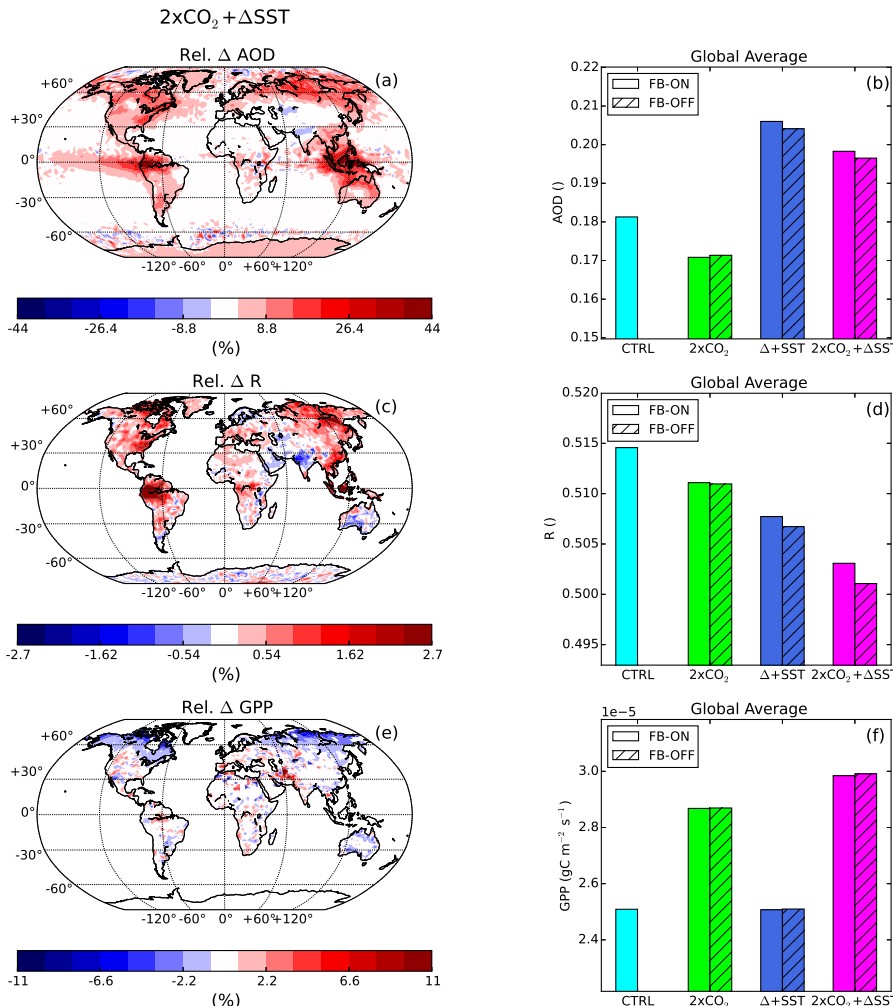

**Figure 9.** The relative difference between the FB-ON and FB-OFF simulations in the annually average AOD (a), R (c) and GPP (e) for the $2xCO_2+\Delta SST$ experiment. In the bar plots (b, d, f), the globally averaged values of the same variables are shown, for the CTRL simulation as well as the three experiments (both FB-ON and FB-OFF simulations).



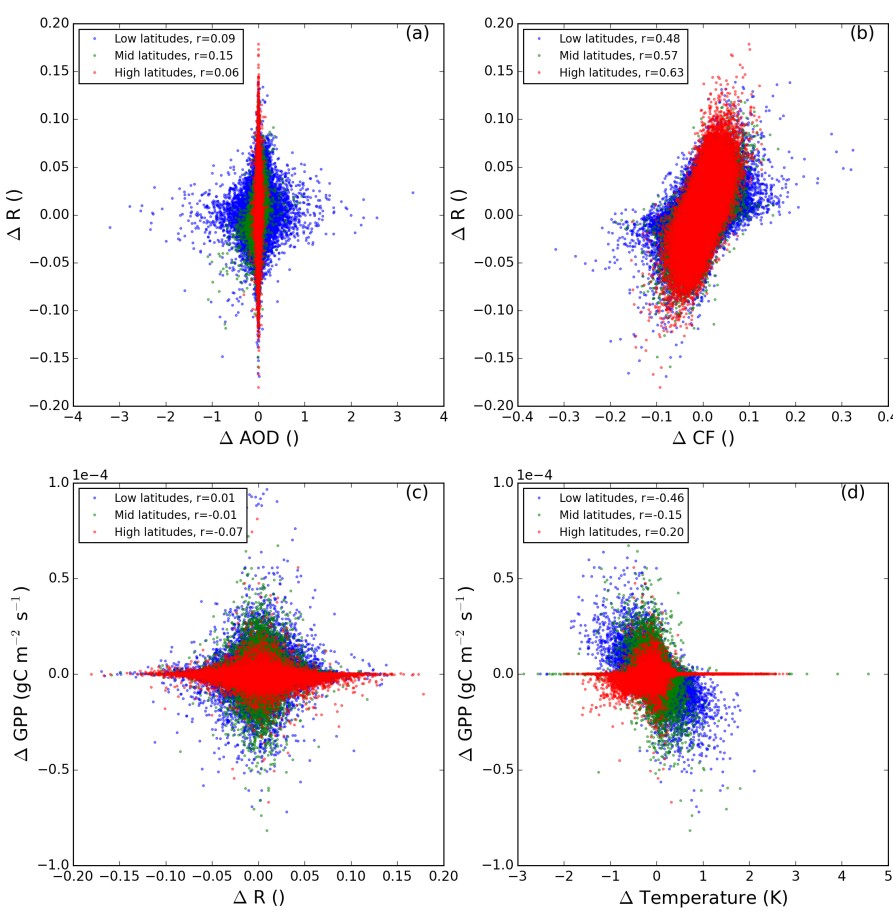

**Figure 10.** Scatter plots of the absolute differences (FB-ON - FB-OFF) in AOD and R in a, CF and R in b, GPP and R in c and GPP and Temperature in the lowest model layer in d. Data from all three experiments ($2xCO_2$, $+\Delta SST$ and $2xCO_2+\Delta SST$) are included. Each dot is a monthly average for one grid box. Only grid boxes with a land fraction of one and GPP greater than zero are included. The dots are coloured according to latitude bands (High latitudes: $55\text{-}90°$, Mid-latitudes: $30\text{-}55°$, Low-latitudes: $0\text{-}30°$) and the correlations coefficient r for each region is show in the legend.





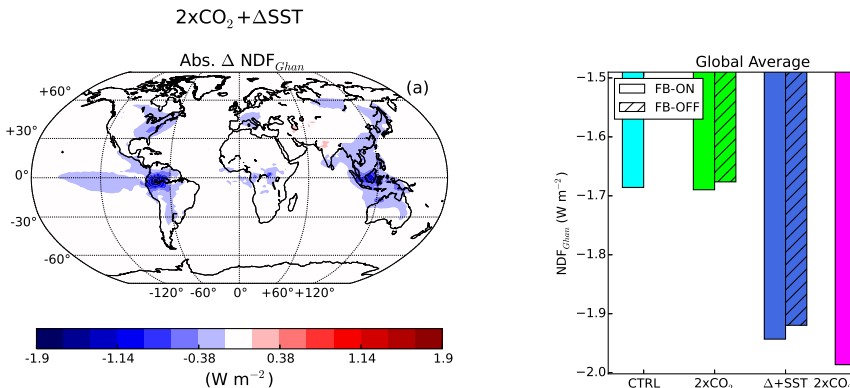

**Figure 11.** The absolute difference between the FB-ON and FB-OFF simulations in the annually average $NDF_{Ghan}$ (a) for the $2xCO_2+\Delta SST$ experiment. In b, the globally averaged $NDF_{Ghan}$ for the CTRL simulation as well as the three experiments (both FB-ON and FB-OFF simulations) are shown.




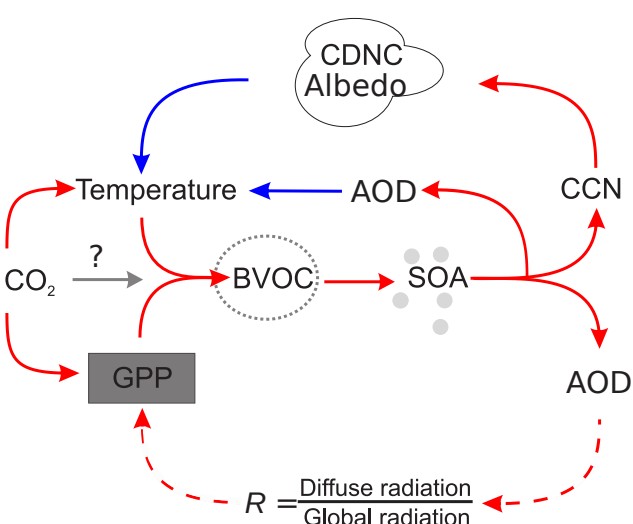

**Figure 12.** Our modified version of the BVOC feedback according to the results from this study. The GPP-branch of the feedback now has dashed lines and the changed AOD has been found to impact temperature.



**Table 1.** Specifications of the $CO_2$ levels, year of the SSTs, BVOC emissions and which meteorology that was used for the nudging for each of the simulations.

| Experiment | $CO_2$ | SSTs and Sea ice | BVOC emissions | aerosol emissions | meteorology |
|---|---|---|---|---|---|
| CTRL | $1xCO_2$ | PD | interactive | PD | CTRL met |
| $2xCO_2$ FB ON | $2xCO_2$ | PD | interactive | PD | $2xCO_2$ met |
| $2xCO_2$ FB OFF | $2xCO_2$ | PD | fixed (CTRL) | PD | $2xCO_2$ met |
| $+\Delta$SST FB ON | $1xCO_2$ | 2080 | interactive | PD | $+\Delta$SST met |
| $+\Delta$SST FB OFF | $1xCO_2$ | 2080 | fixed (CTRL) | PD | $+\Delta$SST met |
| $2xCO_2+\Delta$SST FB ON | $2xCO_2$ | 2080 | interactive | PD | $2xCO_2+\Delta$SST met |
| $2xCO_2+\Delta$SST FB OFF | $2xCO_2$ | 2080 | fixed (CTRL) | PD | $2xCO_2+\Delta$SST met |
| $2xCO_2+\Delta$SST FB ON LA | $2xCO_2$ | 2080 | interactive | PI | $2xCO_2+\Delta$SST met LA |
| $2xCO_2+\Delta$SST FB OFF LA | $2xCO_2$ | 2080 | fixed (CTRL) | PI | $2xCO_2+\Delta$SST met LA |



**Table 2.** Difference in the global annual average $NCF_{Ghan}$, $NDF_{Ghan}$ and total aerosol forcing $TAF_{Ghan}$ between the FB-ON and FB-OFF simulations.

| Experiments | $\Delta NCF_{Ghan}$ (W m$^{-2}$) | $\Delta NDF_{Ghan}$ (W m$^{-2}$) | $\Delta TAF_{Ghan}$ (W m$^{-2}$) |
|---|---|---|---|
| 2xCO$_2$ | -0.11 | -0.014 | -0.12 |
| +$\Delta$SST | -0.19 | -0.025 | -0.22 |
| 2xCO$_2$+$\Delta$SST | -0.43 | -0.058 | -0.49 |
| 2xCO$_2$+$\Delta$SST LA | -0.66 | -0.074 | -0.73 |