# Peer review of "BVOC-aerosol-climate feedbacks investigated using NorESM"

_Atmospheric Chemistry and Physics, 2018_

## Referee Comment (RC1) · Anonymous Referee #1 · 12 Dec 2018

GENERAL COMMENTS:

The authors use the Norwegian Earth System Model (NorESM) to explore the impact of higher temperatures, and carbon dioxide concentrations, on BVOC emissions and subsequent SOA formation. The ESM is used to explore two branches of the feedback between climate and BVOC, via both temperature and gross primary productivity (GPP).

The authors find that, in NorESM, the impact of the temperature feedback on climate (via changing BVOC emissions) is greater than the impact of the GPP-driven feedback on climate (via changing BVOC emissions).

The paper is interesting, within the scope of ACP and suitable for publication following

minor revisions and clarifications:

- It would be useful to understand these results in the context of the other studies you cite in the Introduction (see Specific Comment about that below). The NorESM BVOC feedback seems like it is stronger than that produced by other models but it is difficult to tell. It would be useful if you could include a map (in the Supplementary Material) to demonstrate the temperature increase that is applied in the $\Delta$SST experiment, this would help with interpretation of the changes to BVOC emissions and subsequent feedback strength.

- How is the aerosol hygroscopicity calculated? The impact of a change in the hygroscopicity is mentioned a few times but it's not clear how this is done, and therefore what the impact of changes to the amount of SOA should be

- It would be useful to clarify what you are referring to as cloud forcing (beyond citing Ghan 2013) – what does this include? (i.e. what does and doesn't change in your ESM) Page 9, line 28-29 is confusing – what does this -0.43 W/m2 represent? I think it's the difference between FB-ON and FB-OFF but that is not clear.

SPECIFIC COMMENTS:

- P3, line 17-22: this is a bit confusing because the first two values are feedback terms in W/m2 per K, whereas the second two are radiative effects / radiative forcing values – do they both relate to a doubling of monoterpene emissions? Should we be able to compare all four values, or not?

- P5, line 23: can you be more specific than 'PI levels', sometimes this means 1750, sometimes 1850. What anthropogenic emissions do you use in the other 3 experiments?

- P7, line 12-14: it's not clear why a reduction in LAI would lower the albedo? Increased tree mortality and reduction of LAI would surely increase the albedo of the surface?

- P8, line 9-11: this sentence is confusing, could you rephrase?

- P11, line 8: is there a Fig 11 c?

- P11, line 17: why aren't the emissions exactly the same? Because of the impact of the anthropogenic aerosol (or lack of) on climate (and therefore BVOC emissions) – does this confuse the response?

- P13, line 24-26: can you use any previous literature to comment on the relative strengths of these impacts and the implications for the feedbacks you calculate here? Some studies have found the gas-phase and particle-phase impacts of changes to BVOC emissions to be quite finely balanced (e.g., Unger 2014; Scott et al., 2018)

- P14, line 9: is the right value, should it be 0.49 W/m2?

TECHNICAL CORRECTIONS (there are quite a few typos, these are some I spotted):

- P1, line 20: missing "a" ?

- P3, line 12: change 'boosts' to 'boost'

- P9, line 18: change 'patters' to 'patterns'

- P11, line 22: 'were' = 'where'

- P12, line 18: 'is' = 'in'?

- P13, line 22-23: change 'oxidization' to 'oxidation'

- P21, Figure 1 caption: 'if' should be 'is' or 'of' ?

REFERENCES:

Scott et al., 2018, Nature Communications, 9.

Unger, 2014, Nature Climate Change, 4.

---

## Referee Comment (RC2) · Anonymous Referee #2 · 9 Jan 2019

The authors use the NorESM model to investigate the effects of two different aerosol-based feedback paths on a changing climate. The paper is clear and interesting, falls within the remit of ACP, and should be published after the following minor revisions and clarifications have been addressed:

**1  General Comments**

- Make it clearer that this scenario is an absolute upper limit of response rather than a prediction. While the statement is made, it should be stronger and clearer.

- Make sure all acronyms are defined clearly at their first use.

- Whenever locations of largest absolute/relative changes are mentioned, try to add a line on the expected behaviour resulting from the feedback.

- How were CCN calculated at 0.2 and 1%? Was it based solely on size or was hygroscopicity also included?

- The suggestions for further research in the limitations and uncertainties section was clear and helpful, and provided good guidance for future work.

**2  Specific Points**

P3, L24: How significant is the correlation? Can you perhaps quote a p-value?

P3, L27: "contribute with a global BVOC emissions gain by 1.07" This statement is unclear.

P4, L30-32: In the limitations and uncertainties section, you mention that the findings are sensitive to the choice of nucleation scheme; why was this scheme chosen? Are there are specific issues that may result from the choice?

P7, L27: Is reduction in hygroscopicity included in the model even if it is not shown?

P8, L19: Amount should be number (grammar issue, relating to countability of nouns)

P8, L23: Are activated cloud droplets stored in the model output? Is it possible to compare CCN and activated cloud droplets together?

P8, L29: Are the vertically averaged CDNC weighted somehow?

P9, L2: Is there any albedo calculation to go with the effective radius and cloud water path?

P9, L21: 11 W m$^{-2}$ seems quite high at first glance - can you offer a little context for the

magnitude of this effect and what other effects might cancel it out to reach a sensible climate perturbation?

P9, L24: $\Delta NCF_s$ might be a clearer notation than NCFS.

P10, L1: Do we believe this feedback is already present in the real Arctic atmosphere? Perhaps include a line or two on the potential implications.

P10, L14-19: r = 0.53 is *higher* than 0.08 but it's still not very high...

P10, L24-25: Does total radiation decrease much? Can we maybe see a map of independent behaviour of total and diffuse radiation as well as R?

P11, L9: There is no Figure 11 (c).

P14, L11-12: A 20% reduction in forcing for lower anthropogenic aerosols makes me wonder if you have accounted for the difference between aerosol effects (change in aerosol behaviour at the same time) and aerosol forcings (change compared to pre-industrial effects). Since you do include PI simulations, it's possible that you *do* mean forcings, but I would like to have it be a little clearer.

**3  Figures**

Please explicitly define the red and blue lines in Figures 1 and 12 as increase or decrease in quantity that arrow points at in flow chart.

For Figure 10, it would help to explicitly state the following:

Based on the model output, AOD does not drive diffuse radiation fraction, but cloud fraction does; and diffuse radiation does not drive gross primary product, but temperature does.

Please also include a few lines on whether you expect these relations are because the

model is missing a connecting process and therefore misrepresenting reality, and if so what process might account for the unexpected outcome.

---

## Author Comment (AC1) · 15 Feb 2019

We would like to thank the referees for their helpful comments which will improve this manuscript. We provide the answers to the referee's comments below. The referee's comments are presented in normal font and our responses are given in italic.

**Anonymous Referee #1**

GENERAL COMMENTS:
It would be useful to understand these results in the context of the other studies you cite in the Introduction (see Specific Comment about that below). The NorESM BVOC

feedback seems like it is stronger than that produced by other models but it is difficult to tell. It would be useful if you could include a map (in the Supplementary Material) to demonstrate the temperature increase that is applied in the $\Delta$SST experiment, this would help with interpretation of the changes to BVOC emissions and subsequent feedback strength.

*We have included a map of the difference in the 2 m temperature between the +$\Delta$SST and the CTRL feedback on simulations in the appendix. We choose the 2 m temperature rather than the sea surface temperature increase since it is the 2 m temperature that the vegetation in the model sees, thus it is more directly related to the BVOC emissions. Moreover, the following sentence has been added to the experimental setup section: "The temperature difference over land resulting from the increase in SST is shown in Fig. S1."*

How is the aerosol hygroscopicity calculated? The impact of a change in the hygroscopicity is mentioned a few times but it's not clear how this is done, and therefore what the impact of changes to the amount of SOA should be

*We have added these sentences describing the calculations of the hygroscopicity to the Methods section: "The hygroscopicity of aerosol particles in NorESM is calculated for each "mixture", which is what the background modes are called after they have changed composition and shape through condensation, coagulation and cloud processing. The hygroscopicity is mass-weighted of all component in the mixtures if the particles are uncoated or have thin coating. If the particles have a thick coating ($>$2 nm) the hygroscopicity is instead mass-weighted of the coating itself (Kirkevåg et al. 2018)."*

It would be useful to clarify what you are referring to as cloud forcing (beyond citing Ghan 2013) – what does this include? (i.e. what does and doesn't change in your

[Figure]

ESM) Page 9, line 28-29 is confusing – what does this -0.43 W/m2 represent? I think it's the difference between FB-ON and FB-OFF but that is not clear.

*These 5 sentences have been added to the methods section to better explain the calculations of the cloud forcing: "In this paper, the methods from Ghan (2013) are used to calculate the forcing from clouds and aerosols. The direct radiative forcing ($DRF_{Ghan}$) is calculated as the difference between the net top of the atmosphere radiative flux and the radiative flux neglecting the scattering and absorption of solar radiation by the aerosols ($F_{clean}$). This is calculated in a separate call to the radiation code. Similarly, the net cloud forcing ($NCF_{Ghan}$) is calculated as the difference between $F_{clean}$ and the flux neglecting the scattering and absorption by both clouds and aerosols $F_{clear,clean}$. In the model, the forcings are calculated separately for the short wave and long wave radiation which we have used to calculate the net forcing. " Moreover, we have rewritten the sentence mentioned in the comment to make clear what we mean: "Nevertheless, the difference in yearly global average $NCF_{Ghan}$ is -0.43 W m$^{-2}$ ($SWCF_{Ghan}$ -0.45 W m$^{-2}$, $LWCF_{Ghan}$ 0.02 W m$^{-2}$) between the FB-ON and FB-OFF simulations in the 2xCO$_2$+$\Delta$SST experiment, indicating that the feedback can contribute with a potentially important impact on the future climate on a global scale."*

SPECIFIC COMMENTS:

P3, line 17-22: this is a bit confusing because the first two values are feedback terms in W/m2 per K, whereas the second two are radiative effects / radiative forcing values – do they both relate to a doubling of monoterpene emissions? Should we be able to compare all four values, or not?

*The studies we have cited have used quite different methods and have therefore presented their values as W m$^{-2}$ K$^{-1}$ or W m$^{-2}$. The values cannot be directly*

[Figure]

*compared since the methods used in the studies are very different.*

P5, line 23: can you be more specific than 'PI levels', sometimes this means 1750, sometimes 1850. What anthropogenic emissions do you use in the other 3 experiments?

*By PI levels we mean 1850 levels. We have changed the text to "PI levels (1850)".*

P7, line 12-14: it's not clear why a reduction in LAI would lower the albedo? Increased tree mortality and reduction of LAI would surely increase the albedo of the surface?

*In CLM, the ground (the soil) in this region has a lower albedo than the leaves on the trees. Therefore, the albedo decreases when the LAI is reduced.*

P8, line 9-11: this sentence is confusing, could you rephrase?

*We have changed the sentence from: "The geographical differences in the particle sizes are located further away from the sources than the differences in number concentrations, in particular in the Tropics (not shown)." to "The changes in particle sizes occur further downwind from the sources than the changes in aerosol number concentrations which are more restricted to areas close to the sources, in particular in the Tropics."*

P11, line8: is there a Fig11c?

*No there is not. We have changed this to a.*

P11, line 17: why aren't the emissions exactly the same? Because of the impact of the anthropogenic aerosol (or lack of) on climate (and therefore BVOC emissions) – does

this confuse the response?

*Yes, the emissions are different because of the removal of the cooling from the anthropogenic aerosol. We do not think this confuses the response because this is the same as would be expected from a possible future reduction in the aerosol load.*

P13, line 24-26: can you use any previous literature to comment on the relative strengths of these impacts and the implications for the feedbacks you calculate here? Some studies have found the gas-phase and particle-phase impacts of changes to BVOC emissions to be quite finely balanced (e.g., Unger 2014; Scott et al., 2018)

*We have added these 2 sentences to the discussion in response to this comment: "Previous studies have found BVOC induced changes in the direct aerosol forcing to be roughly balanced by the changes in the forcing from ozone and methane (Unger 2014, Scott et al. 2018). This indicate that part the of forcing (the NDF in this study is 12 % of the total forcing) associated with BVOC feedback investigated in this paper could be offset by changes in ozone and methane lifetime."*

P14, line 9: is the right value, should it be 0.49 W/m2?

*Yes, it should be 0.49 W $m^{-2}$. We have changed this in the text.*

TECHNICAL CORRECTIONS (there are quite a few typos, these are some I spotted):
*We have read through the text once more and removed all the typos we could find.*

P1, line 20: missing "a" ?
*We can't find a missing "a".*

P3, line 12: change 'boosts' to 'boost'

[Figure]

*We have changed the text according to the referee's suggestion.*

P9, line 18: change 'patters' to 'patterns'
*We have changed the text according to the referee's suggestion.*

P11, line 22: 'were' = 'where'
*We have changed the text according to the referee's suggestion.*

P12, line 18: 'is' = 'in'?
*We have changed the text according to the referee's suggestion.*

P13, line 22-23: change 'oxidization' to 'oxidation'
*We have changed "oxidization" to oxidizing*

P21, Figure 1 caption: 'if' should be 'is' or 'of'
*We have changed the text according to the referee's suggestion.*

**Anonymous Referee #2**

GENERAL COMMENTS:

Make it clearer that this scenario is an absolute upper limit of response rather than a prediction. While the statement is made, it should be stronger and clearer.
*We do not quite agree with the reviewer on this. While our temperature increase is at the upper end of possible future scenarios in terms of temperature and $CO_2$ concentrations it is not the absolute upper limit. There are scenarios (such as the new*

[Figure]

*SSP 8.5 scenario) that has higher temperature and $CO_2$ changes than those applied in this study. We believe the statement currently included in the paper is strong enough.*

Make sure all acronyms are defined clearly at their first use.
*We have looked through all the acronyms and made a few corrections.*

Whenever locations of largest absolute/relative changes are mentioned, try to add a line on the expected behaviour resulting from the feedback.
*Out interpretation of the comment is that the referee wants us to add a line on the behaviour expected by the proposed feedback loop when presenting results from each step along the feedback, and we have included this on 7 occurrences in the manuscript.*

How were CCN calculated at 0.2 and 1%? Was it based solely on size or was hygroscopicity also included?
*The CCN calculations include both size and hygroscopicity. The following sentence has been added to the model description section: "Both the size and hygroscopicity of the aerosol particles are used in the calculations of CCN and the activation of aerosols to cloud droplets." We have added a description on how the hygroscopicity is calculated to the manuscript. See also the comment on hygroscopicity by Referee's 1.*

The suggestions for further research in the limitations and uncertainties section was clear and helpful, and provided good guidance for future work.
*We thank the referee for this nice comment.*

SPECIFIC POINTS:

[Figure]

P3, L24: How significant is the correlation? Can you perhaps quote a p-value?
*The investigations by Kulmala et al. investigated the relationships between several parameters with correlations ranging from 0.27 to 0.6 but all with p-values below 0.02.*

P3, L27: "contribute with a global BVOC emissions gain by 1.07" This statement is unclear.
*We have rewritten the statement somewhat to make it clearer: "contribute with a gain in global BVOC emissions by 1.07"*

P4, L30-32: In the limitations and uncertainties section, you mention that the findings are sensitive to the choice of nucleation scheme; why was this scheme chosen? Are there are specific issues that may result from the choice?
*This is the scheme that is included in NorESM. There is development work ongoing testing other schemes in NorESM but we have used the one that is ready and has already been tested and documented. The sensitivity of NorESM to changes in BVOC emissions will be affected by the choice of nucleation scheme but we are not aware of any specific issues that this particular nucleation scheme can cause.*

P7, L27: Is reduction in hygroscopicity included in the model even if it is not shown?
*Yes, reduced hygroscopicity is included in the model.*

P8, L19: Amount should be number (grammar issue, relating to countability of nouns)
*We have changed the text according to the referee's suggestion.*

P8, L23: Are activated cloud droplets stored in the model output? Is it possible to compare CCN and activated cloud droplets together?
*The activated cloud droplets are not stored in the model output.*

[Figure]

P8, L29: Are the vertically averaged CDNC weighted somehow?
*Yes, the vertically averaged CDNC are weighted according to pressure level thickness and cloud fraction.*

P9, L2: Is there any albedo calculation to go with the effective radius and cloud water path?
*Unfortunately, albedo is not an output variable of the model and calculating the albedo from the available output would provide a crude approximation. Therefore, we have refrained from including albedo in the manuscript.*

P9, L21: 11 W m$^{-2}$ seems quite high at first glance - can you offer a little context for the magnitude of this effect and what other effects might cancel it out to reach a sensible climate perturbation?
*The 11 W m$^{-2}$ is the regional forcing during the summer month, which is not all that high. The local yearly average NCF$_{Ganh}$ in this region is up to 2.2 W m$^{-2}$.*

P9, L24: $\Delta$ NCFs might be a clearer notation than NCFS.
*We have changed the text according to the referee's suggestion.*

P10, L1: Do we believe this feedback is already present in the real Arctic atmosphere? Perhaps include a line or two on the potential implications.
*Yes, we believe that the feedback may be present in the Arctic already. However, in this investigation we are not running a coupled model and we are not including other BVOC feedbacks, such as changes methane and ozone lifetime. We can therefore not asses the full effect of the feedback in the Arctic and refrain from speculating on a potential impact from the feedback on the current Arctic climate.*

[Figure]

P10, L14-19: r = 0.53 is higher than 0.08 but it's still not very high...
*We have removed the word much from the sentence.*

P10, L24-25: Does total radiation decrease much? Can we maybe see a map of
independent behaviour of total and diffuse radiation as well as R?
*The total radiation decreases by a maximum of 3.3 % (relative decrease). We have
included a map of the relative difference in total and diffuse radiation as well as r for
the FB-ON and FB-OFF simulations for the $2xCO_2+\Delta SST$ experiment in the supple-
mentary. Half a sentence referring to the figure was also added in the manuscript.*

P11, L9: There is no Figure 11 (c).
*We have changed this to "a".*

P14, L11-12: A 20% reduction in forcing for lower anthropogenic aerosols makes me
wonder if you have accounted for the difference between aerosol effects (change in
aerosol behaviour at the same time) and aerosol forcings (change compared to pre-
industrial effects). Since you do include PI simulations, it's possible that you do mean
forcings, but I would like to have it be a little clearer.
*Yes we mean forcing. We have changed the manuscript to make this clearer and the
sentence now reads: "Thus, the forcing associated with the BVOC feedback could
offset this by about 13 %, or even up to 20 % given a strong reduction in anthropogenic
aerosols."*

FIGURES:

[Figure]

Please explicitly define the red and blue lines in Figures 1 and 12 as increase or decrease in quantity that arrow points at in flow chart.

*We have added this text to both figure captions: "The red arrows in the figure indicate that if the variable at the start of the arrow increase then the variable at the end of the arrow is also expected to increase. A blue arrow at the other hand means that an increase in the variable at the start of the arrow is expected to result in a decrease in the variable at the end of the arrow."*

For Figure 10, it would help to explicitly state the following: Based on the model output, AOD does not drive diffuse radiation fraction, but cloud fraction does; and diffuse radiation does not drive gross primary product, but temperature does.

*We have added the referee's text to caption 10.*

Please also include a few lines on whether you expect these relations are because the model is missing a connecting process and therefore misrepresenting reality, and if so what process might account for the unexpected outcome.

*We do not believe that the model is missing connecting processes since both the effects from aerosols on diffuse radiation and the effect from diffuse radiation on the vegetation is included in the model. There are always uncertainties in parametrizations in the models, but we refrain from speculating on these.*
* * *